 

# A common mechanism underlies changes of mind about decisions and confidence

Ronald van den Berg[1], Kavitha Anandalingam[1], Ariel Zylberberg[2,3,4], Roozbeh Kiani[5], Michael N Shadlen[2,3,4,†], Daniel M Wolpert[1*†]

[1]Computational and Biological Learning Laboratory, Department of Engineering, Cambridge University, Cambridge, United Kingdom; [2]Kavli Institute, Columbia University, New York, United States; [3]Howard Hughes Medical Institute, Columbia University, New York, United States; [4]Department of Neuroscience, Zuckerman Mind Brain Behavior Institute, Columbia University, New York, United States; [5]Center for Neural Science, New York University, New York, United States

**Abstract** Decisions are accompanied by a degree of confidence that a selected option is correct. A sequential sampling framework explains the speed and accuracy of decisions and extends naturally to the confidence that the decision rendered is likely to be correct. However, discrepancies between confidence and accuracy suggest that confidence might be supported by mechanisms dissociated from the decision process. Here we show that this discrepancy can arise naturally because of simple processing delays. When participants were asked to report choice and confidence simultaneously, their confidence, reaction time and a perceptual decision about motion were explained by bounded evidence accumulation. However, we also observed revisions of the initial choice and/or confidence. These changes of mind were explained by a continuation of the mechanism that led to the initial choice. Our findings extend the sequential sampling framework to vacillation about confidence and invites caution in interpreting dissociations between confidence and accuracy.

*For correspondence: wolpert@ eng.cam.ac.uk

†These authors contributed equally to this work

**Competing interests:** The authors declare that no competing interests exist.

## Introduction

Many decisions benefit from the acquisition of multiple samples of evidence acquired sequentially in time. In that case, a decision maker must decide not only about the proposition in question but also about when to terminate deliberation. The ensuing tradeoff between speed and accuracy is explained by *sequential sampling with optional stopping* models in which evidence is accumulated to some stopping criterion or bound (*Link, 1975*; *Ratcliff and Rouder, 1998*). The mechanism receives experimental support from human psychophysics and neural recordings in monkeys and rats (*Gold and Shadlen, 2007*; *Brunton et al., 2013*; *Shadlen and Kiani, 2013*; *Hanks et al., 2015*). The same framework also explains the confidence that a decision is correct (*Kiani and Shadlen, 2009*; *Kiani et al., 2014a*). This is because the quantity that is accumulated, termed a decision variable (DV), when combined with elapsed decision time, maps to the probability that a decision rendered on its value will be correct (*Kiani and Shadlen, 2009*; *Drugowitsch et al., 2014*). The attribution of confidence is important for guiding subsequent decisions, learning from mistakes and exploring alternatives. Thus, when the decision maker terminates deliberation, the choice is accompanied by a degree of certainty (i.e., confidence), based on the same stream of evidence that supported that decision (*Fetsch et al., 2014*).

This last point remains controversial, however, for there are many instances when the confidence in a decision and the decision itself are dissociable. For example, human decision makers tend to overestimate their certainty about choices based on truly ambiguous evidence (*Fischoff et al.,*

**eLife digest** To understand how the brain makes decisions is to understand how we think – how we deal with information, interpret it and agree with a particular interpretation of the information. Neuroscience has begun to uncover the mechanisms that underlie these processes by linking the activity of nerve cells in the brain to different aspects of making decisions. These include how long it takes to reach a decision, why we make errors and how confident we feel about a decision.

Sometimes when we make a decision and have committed to an answer, we then change our minds. Now, van den Berg et al. have asked whether the brain mechanisms that support a change of mind also support a change in confidence. To investigate this problem, human volunteers were asked to perform a difficult task where they had to decide whether a field of randomly moving dots had a tendency to drift to the left or to the right.

During the experiment, van den Berg et al. recorded how long the volunteers took to make their decision, how confident the volunteers felt about their choice, and whether they were correct. Analyzing this data revealed that all of these measures could be explained by a mechanism where the brain accumulates evidence only until there appears to be enough evidence to favor one choice over the other. This process specifies how confident an individual should be based on the quality of the sensory evidence and how long it takes to make a decision.

In addition, van den Berg et al. found that occasionally a volunteer changed their mind about how confident they were about a decision after they'd made it, as if they had continued to think about it. This was despite the volunteers receiving no more information about the task or how well they had done once they had made their decision. Therefore, it appears that the brain processed additional information that had already been detected but did not have time to affect the initial choice.

The activity of the nerve cells in the brain was not recorded as the volunteers made their decisions. Future experiments that incorporate these measurements could help reveal how the brain performs the necessary computations and account for the time delay seen in processing some of the data. Where is this delayed information processed in the brain, and how does it lead to a change of mind?

1982; *Baranski and Petrusic, 1994*; *Erev et al., 1994*; *Drugowitsch et al., 2014*; *Kiani et al., 2014a*), and they can perform above chance level yet report they are guessing (*Kunimoto et al., 2001*). These and other observations have led psychologists to suggest that confidence and choice may be guided by different sources of evidence (*Pleskac and Busemeyer, 2010*; *Zylberberg et al., 2012*; *Moran et al., 2015*), or that the evaluation of the same evidence differs fundamentally in the way that it affects choice and confidence (*Fleming and Dolan, 2012*; *Maniscalco and Lau, 2012*; *De Martino et al., 2013*; *Ratcliff and Starns, 2013*). The latter distinction is captured by the notion of a 1st-order confidence that is based rationally on the evidence in support of the decision and a 2nd-order confidence that can depart from this evidence. As this distinction rests on a proper understanding of the mechanism that supports choice and confidence, it is possible that some of the observations taken as support for higher order explanations of confidence are simply accounted for by deficiencies of the theory of 1st order choices.

Naturally, if a decision maker acquires additional information after committing to a choice, she might wish to revise a decision, or the confidence in that decision, or both. Such changes of mind occur occasionally, even when there appears to be no additional information available after the initial decision has been rendered. The sequential sampling framework (e.g., bounded evidence accumulation) offers a natural account of this phenomenon, because the mechanism incorporates processing delays, which leave open the possibility that the brain might have access to additional evidence that did not influence the initial decision and which might instead influence a revision. Evidence for such a process was adduced to explain reversals of perceptual decisions in humans and monkeys (*Rabbitt, 1966*; *Rabbitt and Vyas, 1981*; *McPeek et al., 2000*; *Caspi and Beutter, 2004*; *Van Zandt and Maldonado-Molina, 2004*; *Resulaj et al., 2009*; *Burk et al., 2014*; *Kiani et al., 2014b*; *Moher and Song, 2014*). Here, we address the possibility that this same mechanism can account for

a revision in the confidence a decision maker assigns to his or her choice. We hypothesized that such revisions might account for an apparent dissociation between degree of confidence and choice.

We asked humans to decide about the net direction of motion in a dynamic random dot display, using a variety of difficulty levels. They simultaneously indicated both their choice and the confidence in that choice by moving a handle with their arm. We show that choice, confidence, and reaction time are explained by a sequential sampling mechanism operating on a common evidence stream. On a small fraction of trials, subjects changed their initial decision about the direction of motion and, more frequently, about their confidence. We show that confidence and choice are informed by the same evidence, both at the initial choice and on any subsequent revision. However, changes of confidence arising through post-decision processing can contribute to an apparent dissociation between confidence and decision.

## Results

Four subjects performed the motion direction discrimination task illustrated in *Figure 1a*. The stimulus duration was controlled by the subjects (*Figure 1b*), who were trained to make fast but accurate decisions (see Materials and methods). Whenever ready, the subject moved a handle from the home position to one of four choice targets, thereby indicating simultaneously the direction decision and confidence level. On most trials subjects made an approximately straight-line movement to one of the choice targets. However, although the random dot stimulus was extinguished on movement onset, on a proportion of trials subjects changed their choice or confidence rating or both, initially reaching towards one target but deviating to reach another (*Figure 1c*). Our central hypothesis is that a common mechanism explains the rate of these occurrences, as well as the speed, accuracy and confidence in the subjects' initial decisions.

### Initial decisions

We wish to account for the effect of stimulus difficulty on the initial direction choice, the confidence level associated with the choice and the time taken to make the decision. Not surprisingly, stronger motion supported more accurate (*Figure 2a*) and faster decisions (*Figure 2b*), that tended to be assigned the higher confidence (*Figure 2c*). The interplay between confidence, accuracy and reaction time can be appreciated from the breakdown of the data within the panels of *Figure 2* (indicated by color). When subjects were more confident, they tended to be more accurate (*Figure 2a*; P<0.0001 for all subjects) and faster (*Figure 2b*; P<0.0001 for all subjects). In addition, subjects reported high confidence more often on correct choices than on errors (*Figure 2c*), and they tended to be more confident on those errors associated with stronger motion (P<1e-5 for 3 subjects and P = 0.50 for S3).

These regularities are consistent with a common mechanism of bounded evidence accumulation to support choice, reaction time and confidence (*Kiani et al., 2014a*). The process involves a race between a mechanism that accumulates evidence for right, and against left, and a process that integrates evidence for left, and against right (*Usher and McClelland, 2001*; *Mazurek et al., 2003*). Since these processes obtain evidence originating in the same stimulus, they will tend to accumulate noisy evidence of opposite sign (i.e., anti-correlated). However, the neural noise associated with the accumulations need not be perfectly anticorrelated, so they are depicted separately (*Figure 3a*). The first process to reach an upper bound determines the choice and decision time.

The state of both the winning and losing processes as well as decision time (*Figure 3b* shows the races on the same plot) confer an expectation that a decision rendered on the evidence is likely to be correct, what we term confidence or belief. Since the winning process is at a fixed bound, the confidence is adequately summarized by the height of this bound, decision time, and the state of the losing accumulator. Thus the degree of belief is based on the balance of evidence (i.e., difference) between winning and losing DVs, as well as the decision time. This model has been tested in neurophysiology and a previous report in humans (*Kiani et al., 2014a*) which extends ideas from signal detection theory, race, and Bayesian models (*Vickers, 1979*; *Kepecs et al., 2008*; *Maniscalco and Lau, 2012*; *Drugowitsch et al., 2014*). Its main distinction is the recognition that the time to decision influences confidence (*Fetsch et al., 2014*; *Kiani et al., 2014a*). For example, this explains why confidence associated with errors increases with stronger coherences: stronger motion induces faster decision times for correct and errors alike. For simplicity, we assumed that

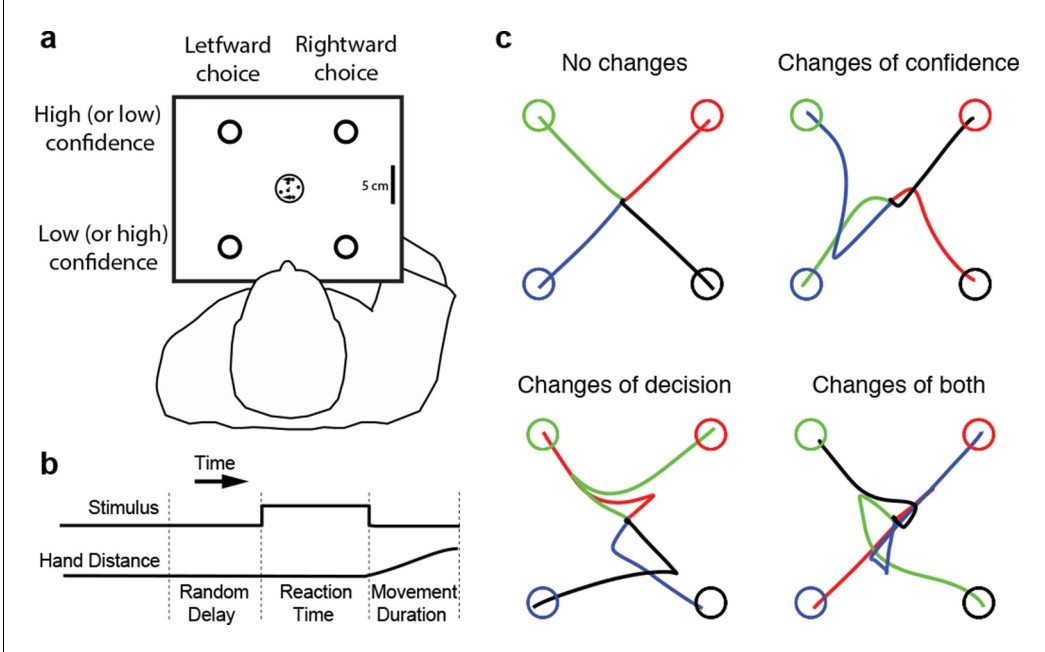

**Figure 1.** Experimental paradigm and sample trajectories. (a) Schematic of the visual display (rectangle). Participants judged the perceived direction of motion of a central random-dot display and whenever ready, moved a handle from the central home position to one of four choice targets, thereby indicating simultaneously the direction decision (leftward vs. rightward targets) and confidence level (top vs. bottom targets). (b) The time course of events that make up a trial. Each trial started when the participant's hand was in the central home position. The subject controlled motion viewing duration, as the motion stimulus was extinguished when the handle left the central homeposition. The trial ended when the participant reached one of the targets. (c) Sample hand trajectories from one participant. Most trajectories extend directly from the central home position to one of the choice targets. In a fraction of trials, the trajectories change course during the movement, indicating a change of confidence, change of direction-decision and occasionally change of both direction-decision and confidence. The trajectory colors indicate the target of their initial choice.

subjects adopt a consistent criterion on "degree of belief" (log odds, *Figure 3b*) to decide in favor of high or low confidence.

The smooth curves in *Figure 2* are fits of the model (4 parameters; see Materials and methods), which capture the main features of the subjects' choices. Naturally, subjects differed in their sensitivity, speed/accuracy and confidence which was accounted for by the parameters of the fits (*Table 1*). They also differed in their criteria for categorizing degree of belief into high and low, which can be appreciated by the separation of choice functions by confidence report. To maximize the number of points, given the reward structure (see Materials and methods), subjects should choose high confidence if they believe the probability of a correct choice is greater than two-thirds, which corresponds to a log-odds of 0.69. Interestingly the inferred criteria for all the subjects were close to this optimal criterion, although all subjects were somewhat risk-averse (probability thresholds of 0.71, 0.78, 0.71 and 0.75; thresholds in log-odd units shown in *Table 1*). There are some noteworthy discrepancies between model and data (e.g., proportion of high confidence choices at 0% coherence for 3 of 4 subjects), but the model captures the trend in the confidence ratings on error trials, mentioned above (3 of 4 subjects; *Figure 2c*), even though the fits themselves are dominated by the more numerous correct choices. The model is vastly superior to two alternative formulations, which would explain confidence on balance of evidence or deliberation time but not both (compared to the original model, log likelihood is reduced by 63–1180 and 2301–3749 across participants for these models, respectively; see Materials and methods).

Importantly, the fits of the initial choices allow us to characterize the state of the DV—including decision time—leading to the subjects' choice and the mapping of this DV to confidence. This is the starting point to address the possibility of revising the initial choice and/or confidence rating



**Figure 2.** Interplay between initial confidence, accuracy, and reaction time. (a) Proportion correct responses as a function of motion coherence split by high (blue) and low (red) confidence decisions. (b) Mean reaction time as a function of motion strength as in (a). (c) Probability of a high confidence initial choice as a function of motion coherence, split by correct (green) and error (magenta) trials. Data are means and s.e.m.; curves are model fits. Only data with 10 or more trials are plotted. For clarity some of the points have been jittered horizontally.

The following figure supplements are available for figure 2:

**Figure supplement 1.** Initial choice behavior for Subject 5.

**Figure supplement 2.** Non-stationary behavior of Subject 4.

afterwards. The critical assumptions are (i) the same information was used to reach an initial choice about direction and confidence, and (ii) there is additional information available to the decision maker after committing to a choice and confidence category, despite the disappearance of the random dot motion upon response initiation.

Both assumptions are supported by the model-free analyses depicted in *Figure 4*. We calculated psychophysical kernels using the information from the stochastic motion displays to determine the time frame over which information in the stimulus affected both components of the choice. To do this we used only weak motion strengths and examined the residual motion energy after removing the means associated with motion direction and strength (see Materials and methods). The traces in *Figure 4a* show the averages of these residuals grouped by choice (*Figure 4a*, top) and grouped by

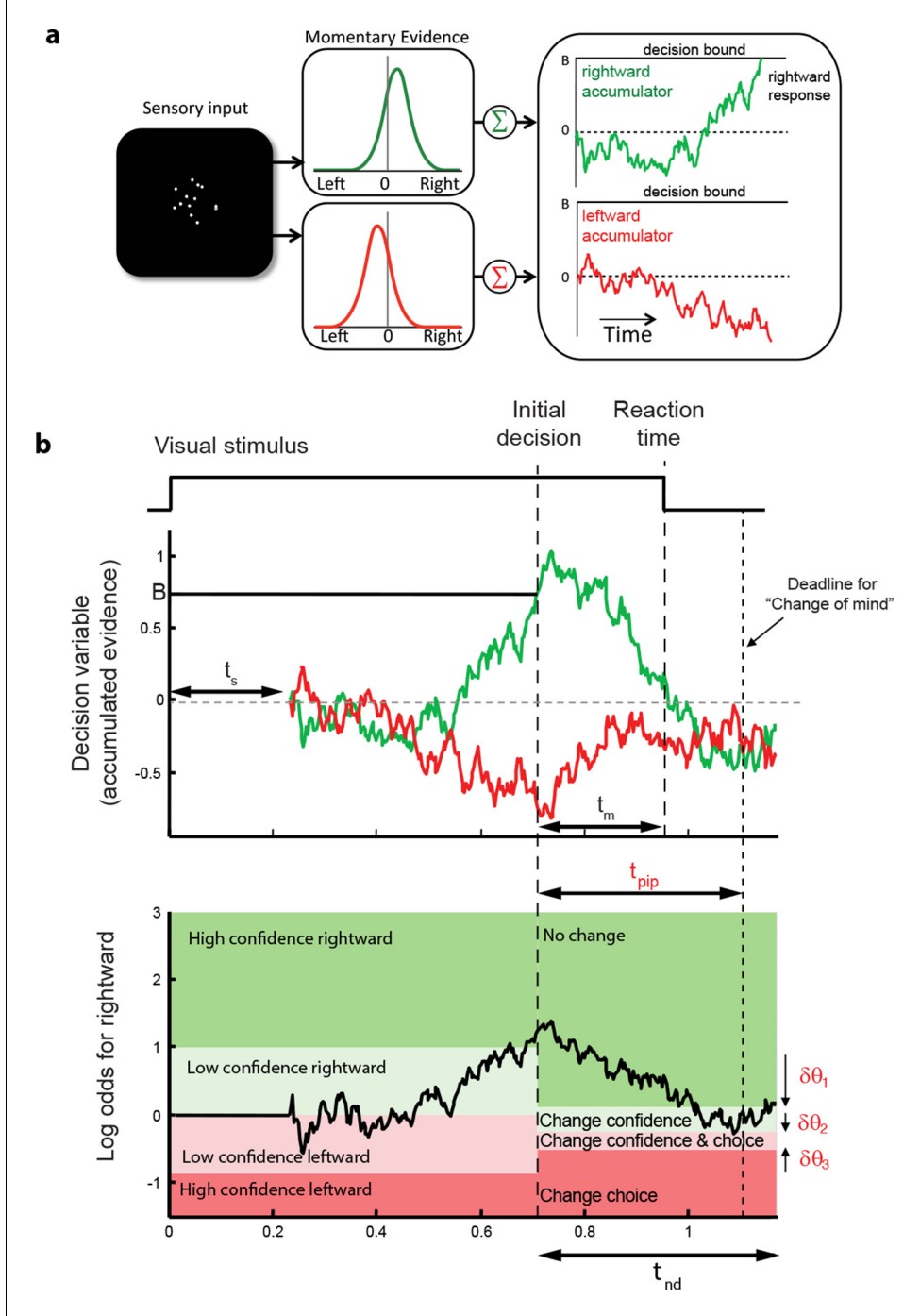

**Figure 3.** Information flow diagram showing visual stimulus and neural events leading to an initial decision, response time, and a possible change of mind. (a) Competing accumulator model of the initial decision. Noisy sensory evidence from the random dot motion supports a race between a mechanism that accumulates evidence for right (and against left) and a mechanism that integrates evidence for left (and against right). Samples of momentary evidence are drawn from two (anti-correlated) Gaussian distributions with opposite means, which depend on the direction and motion strength of the stimulus. In this case the motion is rightward; therefore, the momentary evidence for right has a positive mean, and the rightward accumulator has positive drift. The first accumulation to reach an upper decision bound determines the choice and decision time. In this case the decision is for a rightward response. (b) Evolution of the decision variables (top) and log-odds (bottom) in the task. For simplicity we plot both accumulations on the same graph. At the initial decision time, the state of both the winning and losing processes as well as decision time confer the log-odds that a decision rendered on the

*Figure 3 continued on next page*

*Figure 3 continued*

evidence is likely to be correct, what we term confidence or belief. The bottom plot shows the log-odds of a rightward choice being correct, calculated from the decision variable and time. We assume that subjects adopt a consistent criterion $\theta$ on "degree of belief" to decide in favor of high or low confidence. Note that the decision is terminated by the decision variable (top), not the log-odds (bottom). Although the motion stimulus is displayed up to the reaction time, the decision does not benefit from all of the information, owing to sensory and motor delays ($t_s$ and $t_m$, respectively). In the post-decision period, the accumulation therefore continues. Changes of confidence and/or decision are determined by which region the log-odds is in after processing for an additional time, $t_{pip} \leq t_s + t_m$. To incorporate energetics costs of changing a decision and having to reach mid-movement to a new target we allow the initial bounds to move from their initial levels ($\delta\theta_1$-$\delta\theta_3$). This example uses the parameter fits for Subject 1 and a 3.2% coherence with an initial high confidence correct rightward decision followed by a change of confidence to a rightward, low-confidence decision (for an initial low-confidence example see *Figure 3-figure supplement 1*).

The following figure supplement is available for figure 3:

**Figure supplement 1.** A second example of the evolution of decision variables (top) and log-odds (bottom) in the task.

confidence rating (*Figure 4a*, bottom). The similar time course of the confidence and choice kernels implies that the initial direction and confidence choices were supported by a common evidence stream. Moreover, by discounting the lag and smoothing introduced by the motion filter (inset), it is apparent that subjects relied on information from the beginning of the display until ~400 ms before movement onset (arrows; see Materials and methods and *Figure 4—figure supplement 1*) to guide both their direction choices and their confidence. These values are consistent with the estimates of non-decision time ($t_{nd}$) obtained from the model fits to the initial choices and RTs (*Table 1*). We next evaluate our hypothesis that some of the additional ~400 ms of information, which did not inform the initial direction and confidence choices, affects changes of mind about direction, confidence or both.

## Changes of mind

After indicating their initial choices, subjects occasionally changed the direction of their hand movement to indicate a different direction or level of confidence or both (*Table 2* and *Figure 5*). The frequency of these changes of mind varied between subjects, from 2.0 to 8.8% of trials. Changes of decision were more likely if the initial decision was an error than if it was correct (*Figure 5a*). These corrections were more likely if the motion information was stronger. Of course, errors were less frequent with stronger motion and less frequent than correct responses (*Figure 2a*). For three of the four subjects, changes of decision corrected an initial error more often than they spoiled an initially correct choice (P<0.001, P<0.001, P = 0.084, P<0.001), consistent with previous reports (*Resulaj et al., 2009*; *Burk et al., 2014*).

**Table 1.** Parameter fits for the initial decision parameters and post-initiation processing parameters.

|  |  | Subject 1 | Subject 2 | Subject 3 | Subject 4 |
|---|---|---|---|---|---|
| Initial decision parameters | B | 0.74 | 0.73 | 1.07 | 0.74 |
|  | $\kappa$ | 13.64 | 12.86 | 8.69 | 19.50 |
|  | $\mu_{tnd}$ (s) | 0.461 | 0.421 | 0.409 | 0.427 |
|  | $\theta$ | 0.89 | 1.26 | 0.87 | 1.12 |
| Post-initiation processing parameters | $t_{pip}$ (s) | 0.395 | 0.235 | 0.390 | 0.285 |
|  | $\delta\theta_1$ | 0.77 | 1.32 | 1.16 | 0.79 |
|  | $\delta\theta_2$ | 0.24 | 0.32 | 0.44 | 0.06 |
|  | $\delta\theta_3$ | -0.36 | -0.57 | -0.26 | -0.69 |

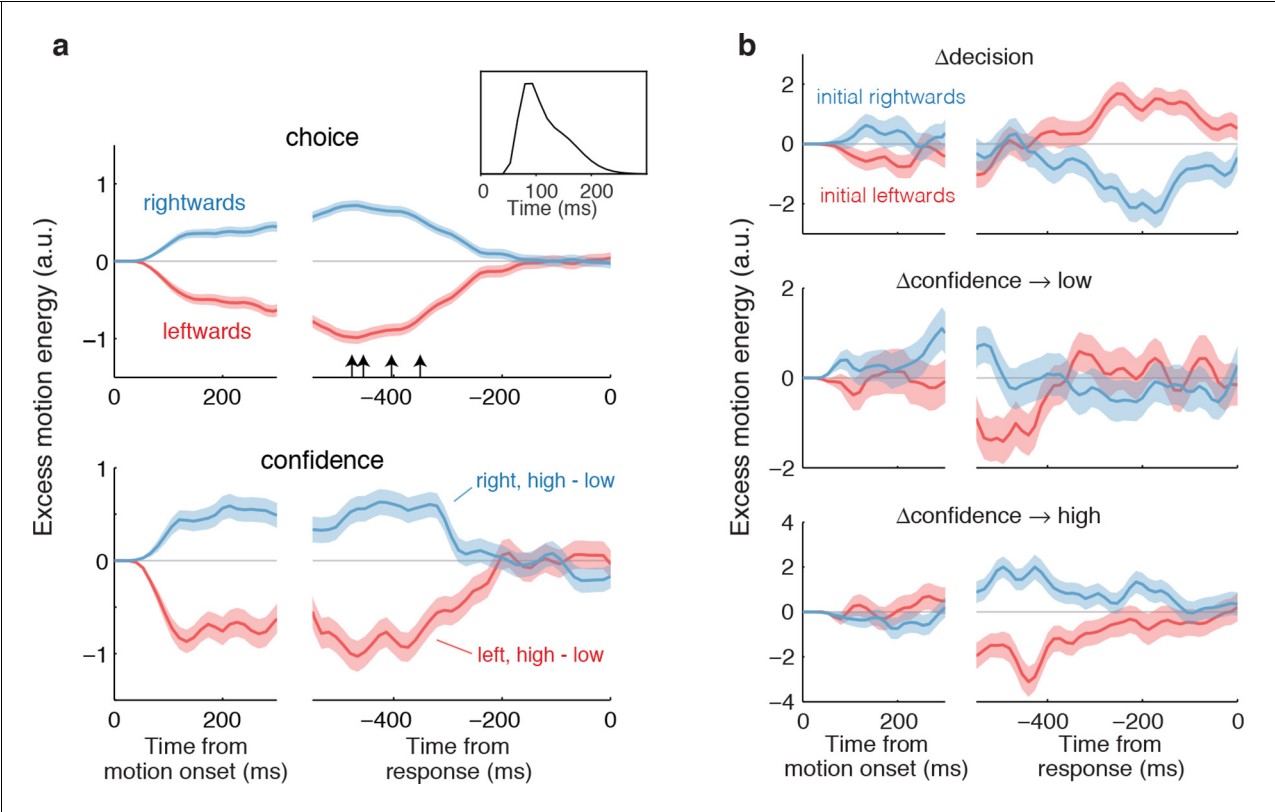

**Figure 4.** Influence of motion information on choice and confidence. (a) Stimulus information supporting initial choice and confidence coincide. Motion-energy residuals were obtained by applying a motion energy filter to the sequence of random dots presented on each trial, and subtracting the mean of all trials having the same coherence and direction of motion. Positive (negative) residuals indicate an excess of rightward (leftward) motion. In each panel, data are aligned to stimulus onset (left) and movement initiation (right). Only motion coherences ≤6.4% are included in the analysis. Inset shows the impulse response of the motion filter to a two-stroke rightward motion "impulse" at $t$ = 0. The upper panel shows the average of the motion energy residuals for rightward (blue) and leftward (red) choices, irrespective of confidence level. Arrows indicate, for each subject, the time prior to the movement initiation when the motion energy fluctuations cease to affect choice. The estimates correct for the delays of the filter (see *Figure 4—figure supplement 1*). The lower panel shows the difference in motion energy residuals between high and low confidence, for each direction choice. Shading indicates s.e.m. (b) Influence of motion energy residuals on changes of mind about direction and confidence. When subjects changed their initial decision about direction (top panel), motion information changed sign just before movement initiation. When confidence changed from high to low (middle panel), residuals were positive or negative for the two direction choices, respectively, and attenuated or reversed sign just before movement initiation. In contrast, late information provided additional support for the initial choice when confidence changed from low to high (bottom panel).

The following figure supplement is available for figure 4:

**Figure supplement 1.** Estimation of the non-decision times from the psychophysical kernels.

The novel insights from the present study derive from the changes in the confidence ratings (*Figure 5b*). For all subjects, changes of confidence were more frequent than changes of decision (*Table 2* and *Figure 5c*; p<1e-4 for all participants). Further, changes of confidence were more likely if the initial decision was low confidence than if it was high confidence (*Figure 5b*). These changes to high confidence were more likely when the initial low confidence accompanied a stronger motion. Note that *Figure 5a* shows the conditional probability of changing the decision about direction, given the initial choice was correct or incorrect (the actual proportions are shown in *Figure 5—figure supplement 1a*; and change of confidence shown in *Figure 5—figure supplement 1b*).

Changes of mind were beneficial to the participants in that on changes of mind trials (either decision, confidence or both) the gain in points over not changing one's mind ranged from 0.72–1.12 points across the subjects (*Table 2*). Changes of both decision and confidence were less common (0.4–1.6%), which may be partly due to an energy cost associated with crossing the workspace (*Burk et al., 2014*; *Moher and Song, 2014*). Those double changes that did occur tended to move

**Table 2.** Pattern of changes of mind for each subject. Total trials performed with percentage of trials for different types of changes of mind. The average additional points earned is the difference in the points earned on change of minds trials compared to those that would have been earned had the subject not changed their mind, divided by the total number of change of mind trials.

| Subject | Total trials | %Trials | | | | Average additional points earned per trial with a change |
| --- | --- | --- | --- | --- | --- | --- |
| | | $\Delta$confidence only | $\Delta$decision only | $\Delta$confidence & $\Delta$decision | All changes | |
| 1 | 9022 | 4.97 | 2.57 | 1.30 | 8.83 | 1.12 |
| 2 | 9023 | 2.78 | 1.31 | 0.40 | 4.49 | 0.90 |
| 3 | 9018 | 1.38 | 0.49 | 0.17 | 2.03 | 0.72 |
| 4 | 5000 | 4.26 | 2.40 | 1.62 | 8.28 | 0.90 |

from high to low confidence (1.03, 0.35, 0.12, 1.36%) compared to trials from low to high (0.26, 0.04, 0.04, 0.26%; p<0.0001 for three subjects and p<0.05 for S3).

We hypothesized that a change of confidence, like a change of decision, can be explained by continuation of the processing of visual information that arrived after the subject had committed to her initial choice (**Figure 3b**). We first evaluated this hypothesis by elaborating the model-free analysis of motion energy (**Figure 4**), described above. This analysis implies that ~400 ms of stimulus information, which did not affect the initial decision, might be available to revise the initial decision about confidence and direction. Indeed, when subjects changed their confidence rating from low to high, the motion information in the post-decision period supported the initial choice (**Figure 4b**, bottom), whereas changes from high to low confidence were associated with motion information in support of the direction opposite to the one chosen (**Figure 4b**, middle), and this trend was amplified for changes of mind about direction (**Figure 4b**, top).

A simple extension of the bounded accumulation model explains the frequency of these changes as well as their dependency on features of the stimulus and the subject's initial report. The model scheme is illustrated in **Figure 3b**. We assumed that evidence was accumulated past the point of the initial decision for a fixed amount of time (a free parameter, $t_{pip}$) that is less than the non-decision time. The state of the belief at $t_{pip}$ determines the final confidence and choice (see Materials and methods). As shown in **Figure 3b**, we allowed for the possibility that subjects might not apply the identical criteria for confidence and choice in the pre- and post-initiation epoch (the $\delta\theta$ parameters; see Materials and methods). The settings of $t_{pip}$ and the $\delta\theta$ parameters capture aspects of the energetic costs, by suppressing changes that would occur with small fluctuations in the evidence or very late in the movement. The trace in **Figure 3b** illustrates an initial high-confidence, rightward choice. Evidence that arrived in the post-initiation period (i.e., too late to affect the initial choice) tended to favor leftward, detracting enough from the belief to support a change of confidence but not enough to support a change of decision in favor of leftward. The example shows a resistance to change because the total evidence actually favors leftward, but the model asserts a change of decision bound that is somewhat below the neutral evidence level.

The curves in **Figure 5** are fits of the model, which capture the main features of the subjects' revisions of both choice and confidence. Note that all parameters of these fits are fixed from the fits to the initial choice and confidence, except for the three $\delta\theta$ parameters and $t_{pip}$ (fit values in **Table 1**). The variations in the participants' behavior are accounted for by the different settings of the model parameters, which are illustrated graphically in **Figure 6** for an initial low and initial high confidence rightward decision. Although the relation between the precise values of the $\delta\theta$ parameters and behavior are hard to intuit, the trends are consistent. After an initial low confidence decision (**Figure 6**, top row), all subjects required more belief to change to a high confidence decision than would have been required for an initial high confidence decision ($\delta\theta_1>0$). Similar hysteresis is seen for initial high confidence decisions (**Figure 6**, bottom row). Moreover, all subjects required more evidence to change their initial decision about direction than a simple reversal in sign of the evidence ($\delta\theta_2>0$). Finally, all participants relaxed their belief criterion when they changed their direction decision while either maintaining or changing to high confidence ($\delta\theta_3<0$). Because most initial choices were high confidence, this strategy would reduce motor effort by avoiding changes of both



**Figure 5.** Changes of confidence and decision. (a) Probability of changes of decision when the initial decision was an error (dark red) or correct (light red) as a function of motion strength. Circles show subject data (mean ± s.e.m) and curves are model fits. (b) Probability of changes of confidence when the initial decision was low confidence (dark blue) or high confidence (light blue) as a function of motion strength. (c) Proportion of trials with changes of confidence (blue) and changes of decision (red) as a function of motion strength. These predictions (curves) are evaluated only at the motion strengths that were presented to the subjects, because they were obtained by using the fits from (a and b) together with the proportion of actual initial choices (error/correct and high/low confidence) for each motion strength. *Figure 5—figure supplement 1* shows the empirical and model fit proportion of trials corresponding to panels and (b).

The following figure supplement is available for figure 5:

**Figure supplement 1.** Proportion of trials with a change of decision or confidence.

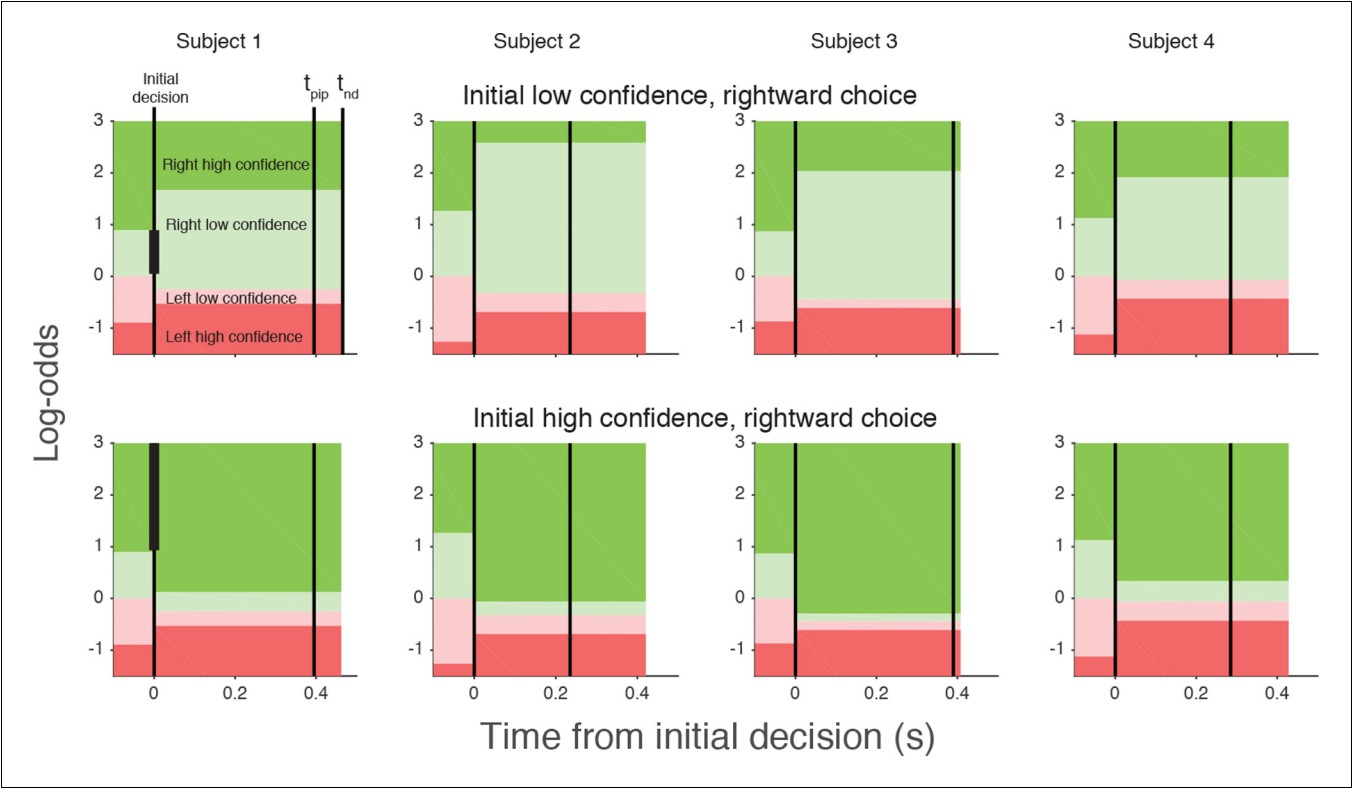

**Figure 6.** Log-odds thresholds for initial decisions and changes of mind for each participant. For initial decisions, the confidence threshold ( $\pm \theta$ for $t<0$, marked by the boundaries between light and dark colors) determines whether the decision has high or low confidence. The initial direction decision is consistent with the sign of the log-odds (rightward for green and leftward for red). Rows show the thresholds for an initial low- (top) and high- (bottom) confidence rightward decision (i.e., initial decisions that end in the region indicated by the vertical black bar in the plot for Subject 1). Thresholds in the post-initiation stage (right of the initial decision line) tend to "move away" from the initial log-odds threshold, consistent with resistance to change. For all participants, the opposite-choice, high-confidence threshold "moves towards" the initial decision, thereby reducing the width of the pink zone. Because initial confidence was more often high (lower row), the narrow pink zone can be interpreted as resistance to a double change of mind about both direction and confidence. Note that the model parameterization requires the pink and red regions to be same for both initial high and low confidence decisions. The non-decision time ($t_{nd}$) and time of post-initiation processing ($t_{pip}$) are also shown for each participant (labels in left top graph).

direction and confidence, as this requires crossing the workspace with a complete reversal of the initial movement.

We wish to emphasize that confidence and direction reports remain coupled via a common mechanism at the time of both the initial and final decision. The observation supports a parsimonious account of decision confidence which is somewhat at odds with recent literature on meta-cognition (see Discussion). Evidence for a meta-cognitive process is adduced from a dissociation between the signal-to-noise properties of the evidence that would support a level of choice accuracy versus the level associated with a degree of confidence (e.g., meta-d′; see *Fleming and Lau, 2014*; *Maniscalco and Lau, 2012*). The meta-d′ statistic is not well behaved in the sequential sampling framework, but the basic logic remains applicable (see Materials and methods). In *Figure 7*, we compare two odds ratios (OR). Both express the relative probability of a high-confidence rating, given a correct or error choice. Along the abscissa, we base the OR on the initial confidence ratings, whereas on the ordinate, we use the final confidence ratings, thus accommodating the change of confidence. In essence, we are pretending that the subject indicated her choice and subsequently told us her confidence. Clearly, there is a systematic discrepancy between the OR pairs (P<1e-4, sign test). All but one of the points are above the main diagonal, and the difference in OR is statistically reliable for 13 of the 19 individual points (4 subjects × 5 non-zero motion strengths with at least 1 error; P<0.05, bootstrap; see Materials and methods). This discrepancy might be regarded as a

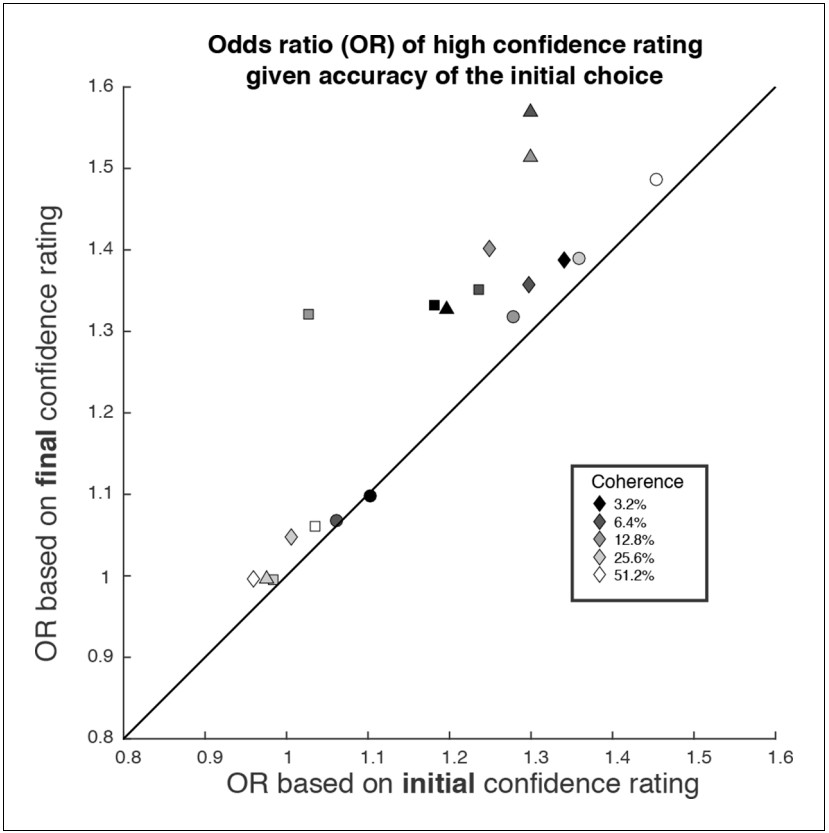

**Figure 7.** The possibility of change of confidence introduces an apparent dissociation between accuracy and confidence. The odds ratio (OR) statistic captures the relative tendency to report high confidence on correct versus error trials. The scatter plot compares ORs calculated from the participants' data using the initial and final confidence report. Both ORs use the correct/error designation for the initial direction decision. The ORs calculated from the initial confidence report establish a baseline: the breakdown of confidence associated with the information that explains the accuracy at each motion strength. The ORs calculated from the final confidence report are larger and might thus be mistakenly interpreted as support for a dissociation between determinants of choice and confidence. Symbol shapes correspond to the four subjects; symbol shading denotes motion strength. An odds ratio can only be calculated if there are error trials (or the ratio is infinite). This necessitated exclusion of one of the points at the highest coherence level.

sign of 2nd-order confidence, whereas it is simply the result of a continuation of the 1st-order process that couples choice and confidence, via bounded evidence accumulation.

## Discussion

Amongst the three manifestations of choice behavior, confidence is perhaps the most important and the least understood—compared to speed and accuracy—because in the world outside the laboratory, we do not always receive immediate feedback about our choices. Often, all we know about the accuracy of a choice is a degree of confidence. The attribution of confidence is important for guiding subsequent decisions, learning from mistakes and exploring alternatives. The present study establishes that this assessment evolves in time and, like the choice itself, can undergo revision with additional evidence. The study of perceptual decision-making offers insight into the process because the stream of evidence can be controlled experimentally. This is especially so in the present study using random dot kinematograms because the temporal stream of evidence is effectively a sequence of independent, statistically stationary samples. In other types of decisions—and most perceptual ones as well—the samples of evidence are derived from internal evaluations and memory processes, which are less well characterized, but which also evolve as a function of time.

When the timing of a choice is under the control of the decision maker, a common strategy is to terminate deliberation upon a sufficient level of evidence. The class of bounded sequential sampling models, including many variants (*Wald, 1947*; *Stone, 1960*; *Laming, 1968*; *Link, 1975*; *Good, 1979*; *Luce, 1986*; *Ratcliff and Rouder, 1998*) explains the tradeoff between speed and accuracy, and many of the essential steps have correlates in neurophysiology (*Romo et al., 2002*; *Heekeren et al., 2004*; *Ploran et al., 2007*; *Heitz and Schall, 2012*; *White et al., 2012*; *Kelly and O'Connell, 2013*; *Hanks et al., 2014*; *Hanks et al., 2015*). The mechanism (and models) also expose a discrepancy between the information supplied to the decision maker and the information that is actually used to make the decision. Specifically, there are processing delays between the arrival of information from the environment and updating its representation in working memory as a decision variable bearing on a proposition, and there are delays between commitment to a decision and communicating this decision through the motor system. Even in relatively fast perceptual decisions, this so-called non-decision time typically exceeds 300 ms.

In previous studies, we showed that the motor system receives a continuous updating of the state of the evidence leading to a choice (*Gold and Shadlen, 2000*; *Selen et al., 2012*) and can utilize the late arriving information to revise an initial choice by continuing the process of deliberation (*Resulaj et al., 2009*; *Burk et al., 2014*). The present study extends these observations to the meta-cognitive operation of assigning a degree of belief that the decision is correct. The finding is important for several reasons. First, it establishes that deliberation in the post-initiation epoch is as rich a process as the deliberation preceding initiation. This implies that termination has a kind of specificity; it marks the end of deliberation for purposes of response initiation, but it does not actually terminate all deliberation. We suspect that this is just one aspect of a more general property. For example, the same information can bear on a variety of potential actions and cognitive operations, which need not share identical speed-accuracy tradeoffs and thus deliberate for different amounts of time.

Second, the finding explains a potential dissociation between the information that decision makers might use to guide confidence and choice. We explain the initial choice and confidence using the identical stream of information, as in a previous study (*Kiani et al., 2014a*). The model is by no means perfect (e.g., it overestimates confidence at the lowest motion strength), but it demonstrates, nonetheless, that on many trials, the final confidence judgments are clearly based on additional information that did not affect the initial choice. This assertion is supported by the analysis of motion energy leading to the confidence choices (*Figure 4*). Just a few hundred milliseconds of additional evidence led to changes in the confidence associated with the initial choice, ranging from 1.5–6.3% of trials across our subjects, and this was enough to induce a clear departure from the level of confidence that ought to be associated with the evidence available at the time of the initial choice. In other words, had we asked our subjects only for their confidence rating after they indicated their initial choice—without an opportunity to revise those choices—we would have detected a different relationship between confidence and accuracy (*Figure 7*). When confidence is assessed after the initial choice, as it is in most experiments, then it ought to come as no surprise that choice and confidence are explained by only partially overlapping sources of information. Importantly, however, this does not imply a dissociation between confidence and the decision (*Pleskac and Busemeyer, 2010*), as the latter can also be revised based on the same additional information.

Of course, in many settings, it is possible that a decision maker might acquire new information after an initial choice, either from the environment or from memory (e.g., reconsidering the evidence, weights and costs), and this could lead to a divergence of choice and confidence. However, our result invites caution when interpreting such divergence as indicators of metacognitive processes. Does the divergence necessarily implicate a process of a different nature, as implied by the distinction between 1st and 2nd order processes (*Fleming and Dolan, 2012*)? Or, is it possible that the mechanism responsible for assigning confidence based on the additional evidence is compatible with the one that supports a choice—perhaps a different choice were the decision maker given an opportunity to revise (consistent with the original notion of type-1 and type-2 decisions; *Clarke et al., 1959*; *Galvin et al., 2003*)? The evaluation of additional evidence may explain why confidence ratings are more strongly correlated with accuracy when they are reported with less time pressure (*Yu et al., 2015*), and why some individuals appear to be better than others at discriminating correct from incorrect decisions (e.g. *Ais et al., 2016*). The central question is whether this additional information affects confidence in a manner that is fundamentally different from the process

that would tie these attributes together. In our study, the answer seems to be negative, and we speculate that the interesting aspects of more real-world distinctions involving re-evaluation of evidence using memory, for example, will require better understanding of the way that memory and decision processes interact but not a fundamentally different mechanism for associating confidence with the evidence arising from that process.

We conclude that confidence, choice and reaction time can be understood in a common framework of bounded evidence accumulation. By definition, confidence may be regarded as metacognitive, simply because it is a report about the decision process itself. Yet the operations leading to confidence seem neither mysterious nor dissociable from the decision process (cf. *De Martino et al., 2013*). That said, there are many unknown features of the underlying mechanism. We know little about the establishment of the mapping between belief and the representation of evidence and time. Nor do we know how a criterion is applied to this mapping to render the categorical choices in our study, or in post-decision wagering decisions (*Kiani and Shadlen, 2009*), or in confidence ratings (*Maniscalco and Lau, 2012*; *Fleming and Lau, 2014*; *Kiani et al., 2014a*). An appealing idea is that this also looks like a threshold crossing in the brain with dynamic costs associated with time (e.g., urgency *Thura et al., 2012*; *Drugowitsch et al., 2012*) and dynamic biases (*Hanks et al., 2011*). Presumably, downstream structures that represent confidence-related values such as reward prediction error, must approximate the mapping between decision variable—represented in LIP and other brain areas—and elapsed time (or number of samples) to achieve this. Of course, downstream structures that control the arm must have access to the decision about both direction and confidence to control the initial reach and possibly revise the movement. To explain our findings, we assumed that choice and confidence are related but processed as if separate attributes, via a correspondence between decision variable and belief. It is intriguing to think that the two dimensions, which are bound together into a single action by the motor reach system in our task, could be dissociated in cognition and memory.

## Materials and methods

### Subjects

Six naïve right-handed subjects, between the ages of 21 and 34, participated in this study. The Cambridge Psychology Research Ethics Committee approved the experimental protocol, and subjects gave informed consent. Two of the subjects were excluded from the analyses based on poor task performance (see below).

### Apparatus

Subjects were seated and used their right hand to hold the handle of a vBOT manipulandum that was free to move in the horizontal plane and allowed the recording of the position of the handle at 1000 Hz (*Howard et al., 2009*). Subjects were prevented from seeing their arm by a horizontal mirror that was used to overlay virtual images of a downward facing CRT video display, mounted above the mirror, into the plane of the movement. A headrest ensured a viewing distance of around 40 cm.

### Stimulus

Subjects discriminated the direction of motion in a dynamic random-dot motion stimulus (*Roitman and Shadlen, 2002*) presented within an aperture subtending 5 degrees of visual angle. The dots were displayed for one frame (13.3 ms, 75 Hz refresh) and then three frames later a subset of these dots was displaced in the direction of motion while the rest of the dots were displaced randomly. Thus the positions of the dots in frame four, say, could only be correlated with dots in frames one and/or seven but not with dots in frames two, three, five and six. The dot density was 12.5 dots deg$^{-2}$s$^{-1}$ and displacements were consistent with a motion speed of 5 deg/s. The difficulty of the task was manipulated through the coherence of the stimulus, defined as the probability that each dot would be displaced as opposed to randomly replaced.

### Procedure

*Figure 1a* show a schematic of the experimental setup. A trial began when the subject's hand (displayed to the subject as a 0.5 cm radius red circle) was inside the home region, i.e., within 1 cm of a

grey cross approximately 30 cm in front of their body. After a random delay, sampled from a truncated exponential distribution (range, 0.5–2.0 s; mean, 0.82 s), a dynamic random-dot stimulus appeared at the home position. On each trial, stimulus coherence was selected randomly from the set ± 0, ± 3.2, ± 6.4, ± 12.8, ± 25.6, and ± 51.2%, where negative coherences correspond to leftward motion and positive coherences to rightward motion. The sign on the 0% coherence is arbitrary but determined which direction would be rewarded (see below).

Four circular choice targets with a radius of 1.5 cm were displayed at the corners of a 17 x 17 cm square centered on the home position. The two choice targets on the left corresponded to a leftward motion decision and the two on the right to a rightward motion decision. To encourage participants to also report the confidence in their decision, the two choice targets for each motion direction decision had different payoffs for correct and incorrect choices. One target was low-risk with a reward of 1 point for a correct choice and a loss of 1 point for an incorrect choice. The other target was high-risk with 2 points for a correct choice and a loss of 3 points for an incorrect choice. The designation correct/incorrect was assigned randomly on 0% coherence trials.

Subjects judged the direction of the moving random dots and reached to a choice target when ready. We encouraged them to make quick decisions without sacrificing accuracy. They were free to interpret this instruction as they wished; they received no verbal instruction to aim for any particular speed/accuracy regime. Critically, when the movement was initiated—that is, the hand was more than 1 cm from the central cross—the random-dot stimulus was extinguished. The trial ended when the subject reached one of the four choice targets. The time course of a trial is shown in *Figure 1b*. If the movement had not been initiated within 3 s after stimulus onset, an error message appeared ("Too Slow") and the trial would be repeated later in the session. After each trial, auditory feedback was given with a pleasant chime or a low-pitched tone corresponding to a correct and incorrect choice, respectively, and the number of points earned or lost was displayed on the screen. Subjects were instructed to maximize the number of points per trial. To encourage this, a running sum of the points was displayed at the top of the display in a bar graph.

Each experimental session consisted of four blocks of 180 or 192 trials each (15 or 16 trials of the 12 coherences). We generated 48 stimuli with a rightward motion direction (8 for each of the 6 different coherence values). The leftward stimuli were generated by using the same dot locations but horizontally mirrored about the center of the aperture. This ensured that across the stimuli there was no left-right bias due to the motion energy of the stimuli. In most experiments, we used this "double-pass" procedure so that these 96 stimuli were displayed twice, in a random order. In a given session, the vertical orientation of the targets changed from block to block; in half of the blocks (A), the two high-risk targets were at the top of the display, while in the other half (B), the high-risk targets were at the bottom (*Figure 1a*). A session consisted of four blocks (768 trials) ordered ABBA or BAAB, and this alternated from session to session. Each subject took part in 12 experimental sessions (9024–9216 trials). All subjects received extensive training on the motion task, beginning with variable duration viewing, controlled by the computer and choice-reaction-time testing without confidence categories. Subjects passed to the main experiment when choice and reaction time functions were stable.

We required subjects to have sufficient perceptual skills and motivation to perform the task. One subject was excluded based on poor discrimination performance: at the end of the first training session, this subject still performed at chance level on all coherences. A second subject was excluded because s/he responded with high confidence on 95% of trials (and with 90% high confidence even at 0% coherence). After 5 sessions (3330 trials) we decided to replace this subject. We found that despite the idiosyncrasy in this subject's data, our model can fit their initial choices (*Figure 2—figure supplement 1*). Subject 4 showed a qualitative change in behavior on the second half of their data (after ~5000 trials; *Figure 2—figure supplement 2a*), reporting high confidence on nearly all trials. As no stationary model can account for such nonstationary behavior we included only the first 5000 trials in our analysis. However, our model could still fit the initial choices for the omitted second half of this participant's data (*Figure 2—figure supplement 2b*).

## Analysis

We excluded from analysis any trials with a reaction time less than 150 ms (9 trials). For each trial, the final decision was determined by the choice target reached. To determine whether a change of decision had taken place we calculated the area between the hand's path over the first 1 cm of

movement and the vertical line through the hand's initial starting location (i.e. the line separating the left and right choice targets). A change of decision was reflected in the area on the side opposite to the final choice being greater than 0.1 cm² (*Resulaj et al., 2009*). The same procedure was applied to determine changes of confidence with the area now calculated relative to the horizontal line separating the high and low confidence choice targets. On each trial, we thus obtain the initial and final decision: choice (left/right), confidence (low/high), and reaction time (time between stimulus onset and movement initiation). We show combined data for the two target configurations (high/low confidence targets at top/bottom), having reassured ourselves that the arrangement had no detectable effects on choice accuracy (p≥0.4; Fisher's exact test) and only small effects on reaction time (RT) for 2 subjects (magnitudes<3%, p<0.01; t-test). There was a subtle bias for the bottom targets (50.6, 51.9, 52.5 and 51.4%, respectively), possibly due to kinematic factors, but recall that the orientation was balanced across the experiment.

To examine effects of motion strength on confidence, we fit a logistic model to the probability of reporting high confidence, as a function of absolute value of motion coherence (*C*)

$$P_{high} = [1 + \exp(-b_0 - b_1|C|)]^{-1}$$

where and $b_i$ are fitted coefficients. To examine whether confidence judgments were associated with more accurate choices we fit a logistic model to the direction choice data for each subject where the probability of choosing right is given by

$$P_{right} = [1 + \exp(-b_0 - b_1 C - b_2 I - b_3 IC]^{-1}$$

where *C* is the signed motion strength, *I* is an indicator variable (zero for a low confidence choice and one for a high confidence choice). To test for improved sensitivity (accuracy) with high confidence, we evaluated the null hypothesis ($H_0$: $b_3 \leq 0$). To examine whether confidence judgments were associated with different reaction times we analyzed each subject's reaction time as an ANOVA with categorical factors of unsigned coherence and confidence. All comparisons of event frequency (e.g. changes of mind) were performed with the Fisher exact test.

For the model-free analyses of the time course of motion information on choice and confidence (*Figure 4*), we derived choice and/or confidence conditioned averages of stimulus motion energy (psychophysical kernels). Due to the stochastic nature of the motion stimuli, the strength of motion will vary from trial to trial, and even within a trial. To quantify the fluctuations of motion along the horizontal axis, we convolved the sequence of random dots shown on each trial with a pair of spatio-temporal oriented filters, selective for rightward and leftward motion. The filters were matched to the speed and displacement of the coherently moving dots (see details in *Adelson and Bergen, 1985*; *Kiani et al., 2008*). The results of the convolution were summed across space to yield the motion energy for each direction and as a function of time. The net motion energy was obtained by subtracting leftward from rightward motion energy. To average data across trials, we removed the average motion energy associated with each trial's coherence and direction of motion. Because fluctuations have a stronger impact when motion is weak, only the lowest motion strengths (≤6.4% coh) were included in these analyses.

The influence of motion fluctuations on choice and confidence becomes negligible a few hundred milliseconds before movement onset (*Figure 4a*). To obtain empirical estimates of non-decision time, we fit a function to the psychophysical kernel. The shape of the function *f(t)* was derived assuming that the psychophysical kernels decay slowly when aligned on movement onset (*Figure 4a*, right) because of: (*i*) the trial-to-trial variability in the non-decision time (assumed Gaussian), which gradually reduces the number of trials that contribute to the psychophysical kernel, and (*ii*) the additional smoothing introduced by the impulse response of the motion energy (inset of *Figure 4a*). With these assumptions, *f(t)* becomes:

$$f(t) = \alpha(1 - \Phi(t, \mu_{tnd}, \sigma_{tnd})) * IR(t)$$

where $\alpha$ is a scaling parameter, $\Phi$ represents a cumulative Gaussian distribution with parameters $\mu_{tnd}$ and $\sigma_{tnd}$, IR(t) is the impulse response of the motion filter, and $*$ indicates convolution. The curve-fitting procedure entails fitting $\mu_{tnd}$, $\sigma_{tnd}$ and $\alpha$ to match *f(t)* to the psychophysical kernels (least-square fit). *Figure 4—figure supplement 1* illustrates the fitting procedure and shows best fitting parameters for each subject.

Although we do not make use of meta-$d'$ in this paper, we refer to the concept and therefore provide a brief definition. In signal detection theory (SDT), $d'$ refers to the difference between the means of two standard Normal distributions, $X_1 \sim N(\mu_1, 1)$ and $X_2 \sim N(\mu_2, 1)$, where $X_n$ is a random variable and $N(\mu_1, \sigma)$ is a Normal distribution with mean $\mu_1$ and standard deviation $\sigma$. The two distributions might represent signal-plus-noise and noise-alone or the distributions of firing rates of neurons tuned to opposite directions of motion (e.g., *Britten et al., 1992*). For binary classification we can conceive of a single distribution of the difference between samples of $X_1$ and $X_2$, such that $X_\Delta \sim N(d', \sqrt{2})$, assuming independence. $X_\Delta$ thus represents a DV whose sign identifies the more likely alternative. There is a one-to-one correspondence between proportion correct and $d'$. Applied to binary classification tasks, meta-$d'$ is the value of $d'$ that would support the proportions of high-confidence ratings on error and correct choices, respectively, based on the assumption that the confidence designation is based on comparison of $X_\Delta$ to some arbitrary but fixed criterion. Thus, within the SDT framework, if meta-$d' \neq d'$ then it is not possible to account for the confidence ratings using the same signal:noise relationship that supports the choice accuracy, and this discrepancy has been interpreted as a sign of meta-cognitive confidence (*Maniscalco and Lau, 2012*; *Fleming and Lau, 2014*). However, the SDT framework must be extended to account for RT and choice. Under sequential sampling (e.g., bounded drift diffusion), meta-$d' \neq d'$, even when the choice and confidence are explained by the same decision variable. Hence we pursued a more empirical approach.

To examine the extent to which initial and final confidence are related to the correctness of the initial choice (*Figure 7*), we first calculated the odds of a high confidence rating, for correct and incorrect initial decision. For example,

$$odds(high\ confidence\,|\,correct) = P(high\ confidence\,|\,correct)/P(low\ confidence\,|\,correct).$$

From this we calculated the odds ratio

$$OR = odds(high\ confidence\,|\,correct)/odds(high\ confidence\,|\,error)$$

which indicates whether an event is more likely to occur in the first condition (i.e. more often for correct choices, hence $OR > 1$) or second condition (i.e. more often for errors, $OR < 1$). We compared the $OR$s for the initial and final confidence, both with regard to the initial choice. The $OR$s calculated from the initial confidence report establish a baseline: the breakdown of confidence associated with the information that explains the accuracy at each motion strength, analogous to meta-$d'$ equal to $d'$ in the SDT framework described above. A larger $OR$ for the final vs. initial confidence would indicate that a final high confidence response became more probable for correct choices compared to incorrect initial choices. We used a bootstrap (N = 1000) to evaluate the reliability of the inequalities in $OR$s, depicted in *Figure 7*.

## Model

We fit the initial decision data with a model in which the decision process is a race between two accumulators (*Vickers, 1979*; *Usher and McClelland, 2001*; *Gold and Shadlen, 2007*; *Churchland et al., 2008*), one that accrues momentary evidence for right (and against left; R-L) and another that accrues evidence for left (and against right; L-R). Momentary evidence was modeled as draws from a bivariate Gaussian with a mean that depends on the coherence (*C*) such that drift rate (/s) is $\kappa C$ for the rightward and $-\kappa C$ for the leftward accumulator, giving a mean of ($\kappa C, -\kappa C$). The bivariate Gaussian has a negative covariance $\rho$ which determines the extent to which the two accumulators share noise (for example arising from fluctuations in the stimulus). A decision is made when one of the two races crosses a decision bound $B$. The reaction time is determined by the time to reach the decision bound and an additional non-decision time (e.g., due to sensory and motor latencies) which was modeled as a normal distribution with mean $\mu_{tnd}$ and standard deviation $\sigma_{tnd}$. The state of both the winning and losing race together with decision time map directly to the log-odds of being correct (see *Kiani et al., 2014a*). To model confidence, we included a log-odds threshold $\theta$ which separated high from low confidence judgments. For simplicity, we chose a time-invariant threshold. We have also fit data using time-dependent confidence bounds, which improve the fits for all subjects but does not affect the figures visibly and does not affect our conclusions.

To fit the accuracy, confidence, and reaction time of the initial choices, we determined the number of trials for the 4 possible initial choices (correct/high-confidence, correct/low-confidence, error/high-confidence & error/low-confidence) for each (unsigned) coherence, as well as the corresponding mean reaction times. For any setting of the parameters, the model predicts the probability of each of the 4 possible initial choices and the mean reaction times for each coherence level. To fit model parameters we minimized the negative log likelihood (i.e., cost), using a multinomial distribution for the 4 choice types and Gaussian distribution for the sample mean RTs. For analytic simplicity (see below), we used a flat bound (i.e., stationary rather than collapsing), which does not capture the shape of the RT distributions and the mean RT on error trials (*Drugowitsch et al., 2012*). Therefore we used only sample mean RT and its associated s.e.m. for correct trials (and all 0% coherence trials) in the cost function.

To further reduce the degrees of freedom of the model to four we fixed both the covariance $\rho$ to $-\sqrt{0.5} \approx$ -0.71 and non-decision time standard deviation $\sigma_{\mathrm{tnd}}$ to 60 ms (within the normal range from a previous study (*Burk et al., 2014*) and supported by an analysis of motion energy; see Figure 1—figure supplement 1). This choice of covariance assumes that about half the noise is shared between the two accumulators (i.e. arises from the stimulus) and is within the normal range of fitted covariances (*Kiani et al., 2014a*). Importantly, this choice of covariance also allowed us to use an analytic solution to the race model when fitting. That is, we generalized the method of images used in (*Moreno-Bote, 2010*) which provides analytic solution for covariances of 0 and -0.5 (requiring 3 and 5 images respectively). Increasing the number of images leads to a variety of possible covariances (but not to arbitrary covariances). We chose a covariance of $-\sqrt{0.5}$ which requires the use of only 7 images thereby allowing efficient fitting of the subjects' data. Using this analytic method precluded the use of collapsing decision bounds (and no lower reflecting bounds; cf., *Kiani et al., 2014a*). The fitting was performed for each participant using multiple (30) runs of Matlab's fminsearchbnd with a wide range of different initial parameters settings. The variability across runs was minimal, suggesting that the optimization procedure converged to global maximum.

To model changes of mind, we assumed that once the initial decision had been made, the accumulators continued to integrate information that was not accessible at the time of the initial decision (due to latencies in the sensory and motor system) for a further post-initiation period $t_{\mathrm{pip}}$ (constrained to be less than $t_{\mathrm{nd}}$). A change in confidence or choice would occur if at the end of this period the log odds crossed a confidence or choice threshold, respectively (*Figure 3b*). Since there are motor costs involved in changes of mind, we included additional parameters that could move the confidence and choice thresholds away from their initial values in the post-initiation stage by $\delta\theta_1$, $\delta\theta_2$ and $\delta\theta_3$ (*Figure 3b*). The three thresholds split the final decision into four zones and the specification of these thresholds depended on whether the initial choice was low or high confidence. For an initial high-confidence choice, the three thresholds were $\theta-\delta\theta_1$, $-\delta\theta_2$, and $-\theta-\delta\theta_3$, respectively (see *Figure 3b*). For an initial low-confidence decision, the first threshold was instead $\theta+\delta\theta_1$. To fit the model to each subject's data, for each unsigned coherence we calculated the number of trials corresponding to the 16 possible events that could occur over the trial (4 possible initial choices and 4 possible final choices) and again used maximum likelihood to fit the four free parameters carrying over $\kappa$ and $\theta$ from the initial decision fits.

Note that basing the change of decision about direction on log odds represents a simplification of a process similar to the one for the initial decision, for example, one that would operate on the evolving DVs represented by the competing accumulators. In principle, such a mechanism could terminate post-initiation processing more flexibly, but the frequency of change of mind about choice and confidence is too small to evaluate this possibility.

We also evaluated two alternative models for initial choices only, which differ in how they assign high vs. low confidence. Model 1 exploits a classic idea from signal-detection-theory, and assigns confidence based on a threshold on the balance of evidence (ignoring deliberation time) (*Vickers, 1979*; *Kepecs et al., 2008*; *Kepecs and Mainen, 2012*; *Wei and Wang, 2015*). Model 2, assigns confidence based on threshold on decision time only, thus ignoring the balance of evidence. We fit both models to our participants' data. They have the same number of parameters (4) as the original model. A 4-choice model for the initial choice-confidence decision (based on dynamic programming) is worthy of consideration but we have yet to find a satisfactory account of our data with this approach.

## Acknowledgements

We thank the Wellcome Trust, Human Frontier Science Program, Royal Society (Noreen Murray Professorship in Neurobiology to DMW), Howard Hughes Medical Institute, National Eye Institute Grant EY11378 to MNS, a Sloan Research Fellowship to RK, and a Simons Collaboration on the Global Brain grant to RK.

## Additional information

### Funding

| Funder | Author |
|--------|--------|
| Wellcome Trust | Ronald van den Berg<br>Kavi Anandalingam<br>Daniel M Wolpert |
| Howard Hughes Medical Institute | Ariel Zylberberg<br>Michael N Shadlen |
| Royal Society | Ronald van den Berg<br>Kavi Anandalingam<br>Daniel M Wolpert |
| Human Frontier Science Program | Ronald van den Berg<br>Kavi Anandalingam<br>Michael N Shadlen<br>Daniel M Wolpert |
| National Eye Institute | Ariel Zylberberg<br>Michael N Shadlen |
| Simons Foundation | Roozbeh Kiani |
| Alfred P. Sloan Foundation | Roozbeh Kiani |

The funders had no role in study design, data collection and interpretation, or the decision to submit the work for publication.

### Author contributions

RvdB, KA, DMW, Conception and design, Acquisition of data, Analysis and interpretation of data, Drafting or revising the article; AZ, RK, MNS, Conception and design, Analysis and interpretation of data, Drafting or revising the article

### Author ORCIDs

Ronald van den Berg, http://orcid.org/0000-0001-7353-5960
Michael N Shadlen, http://orcid.org/0000-0002-2002-2210
Daniel M Wolpert, http://orcid.org/0000-0003-2011-2790

### Ethics

Human subjects: The Cambridge Psychology Research Ethics Committee approved the experimental protocol, and subjects gave written informed consent.

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
