## [Decision Letter]

Thank you for submitting your work entitled "A common mechanism underlies changes of mind about decisions and confidence" for consideration by *eLife*. Your article has been reviewed by three peer reviewers, and the evaluation has been overseen by a Reviewing Editor (Timothy Behrens) and David Van Essen as the Senior Editor. One of the three reviewers has agreed to reveal his identity: Mark Goldman.

The reviewers have discussed the reviews with one another and the Reviewing Editor has drafted this decision to help you prepare a revised submission.

Summary:

The paper follows previous papers by the senior authors on decision-making tasks involving an accumulation of evidence, and corresponding models for accuracy, reaction time, and confidence judgments. This paper makes major new contributions to these problems, particularly with regard to the issue of confidence, and whether confidence judgements can be naturally explained in the same framework used to explain accuracy and reaction time. The authors use a very clever experimental paradigm for human psychophysics, in which the direction a lever is moved lets subjects not only indicate their choice (left or right) but also whether this choice has high or low confidence. As a result, both changes of choice and changes of confidence can be seen by changes in hand movement.

All of the reviewers had praise for the manuscript, describing the paradigm as elegant and clever, the paper as insightful and the contribution to the state-of-the-art as important because of the unification into a single framework of decisions and meta-decisions, which have preciously been treated as separate entities. They note that there is clear structure in the data in support of a common driver of these variables such as the model-free analyses of motion energy.

The reviewers, however, suggested that the paper should not be published without some essential revisions (that do not require additional data).

Essential revisions:

First, there were concerns about the extent to which the common mechanism argument was a complete description of the data

Two reviewers raised a related point on these lines.

1) The model fits in Figure 2 show a good fit to accuracy and RT data (panels A and B) but a poorer fit to the proportion of high confidence responses (C), particularly for error trials. E.g. subject 1 has only two (out of 6) error-trial data points situated near the fitted line, and only one data point for subject 2 (part of the issue in realising this is that the purple error trial points are often hidden behind the green correct trial points, it would be useful to jitter these sideways for visualisation). This is the case even though there are several hundred error trials particularly at lower coherences with correspondingly tight error bars on confidence proportions. This discrepancy is particularly apparent at 0% coherence, where subjects 1, 2 and 4 show substantially lower confidence than predicted by the model. I'm not suggesting therefore that there can't be a common mechanism that does a good job of explaining all these data features (e.g. perhaps subjects are inferring on coherence, leading to lower confidence at 0% than predicted by a model that marginalizes coherence?). But it seems reasonable that a putative common mechanism driving choice, RT and confidence should do an equally good job of explaining each feature separately.

2) First, it is not clear to me what data could not be fitted by their model, given multiple fitted parameters per subject. It would be nice to see how a model with a separate "confidence-generating" and "decision-generating" circuit would produce data that would not be fit by their model.

3) I do agree though that using Occam's razor, fitting by one model circuit is preferable to two. However, Figure 2 shows a horrible fit of the model to the 0% coherence data points in a systematic direction across subjects. This suggests a systematic problem with the model, but the authors do not mention this discrepancy.

4) There was agreement in the reviewer discussion that simulation of competing models to demonstrate how the data disconfirm a model that does not rely on a common mechanism would be important if the argument is to shift broader opinion. The reviewers regarded such a demonstration as essential. One suggestion during the discussion was as follows: "Perhaps a simplification of their model and then use of Model comparison techniques to adjust for different numbers of parameters would be a good test to show that a separate computation is needed for confidence as they claim."

Related to these concerns, was a concern about data exclusion.

5) This worry is compounded by the exclusion of data that shows aberrant confidence behaviour. One subject was excluded for a strong high-confidence biasand absence of confidence-accuracy correlation, which is indeed at odds with the proposed mechanism linking choice and confidence, but perhaps not with other models. Similarly the second half of subject 4's data is excluded because they changed the way they reported confidence half way through the session. These exclusions seem to unfairly skew the playing field in favour of the authors' single-mechanism account, particularly as the remaining subjects' data constitute only a little over half of all data that was originally collected.

During the discussion it was made clear that although it is understandable that subjects may have effort-related or other biases that make the data uninteresting, it is a concern when this is 2 of 6 subjects, and it is a concern that the particular exclusions seem to act in your favour. Because of this, it was thought to be essential that the excluded data were presented in the supplement, and that the reason for the odd behaviour was investigated and described.

6) There was also an important question about perceived asymmetries in model that the reviewers required clarification about. It was unclear why you switch between working in accumulator space and log-odds space. In the discussion, it was agreed that it might be possible for you to clarify in the text if there was a strong and compelling reason, but if not then it would be interesting to see a symmetric model.

My largest comment is that, while confidence and decisions are clearly put together in a single framework, I found the framework to be a little awkward and/or unclear in what information/type of model informed decisions versus what informed confidence. The authors use a classic "race to bound" model to inform the decision. However, then they appear to use a separate probabilistic judgment (log odds, or maybe log odds ratio-this was a bit unclear) for confidence and for changes of decision and confidence. This seems awkward: why not just use the log odds metric for all stages of the task instead of artificially making a race to threshold for two completely independent integrators for the decision and then a separate probabilistic framework for confidence and changes of decision?

Or, maybe I am misinterpreting the log odds graph and this could simply be interpreted as a one-to-one mapping between the decision variable and the log odds (so that one could indeed reformulate the race to bound in the decision variable as a race to bound in log odds). If this is the case, then I find the log odds measure for change of choice to be somewhat strange: in the race to threshold model, there are two completely independent accumulators that are racing against each other. In the log odds analysis, it would seem very awkward if only the odds for the initially winning accumulator was used (i.e. I'm assuming either the authors are using odds or odds ratios based upon using both accumulators, or they are just using the winning accumulator, since the figures only indicate a single trace in log odds). If only the winning accumulator is used, then it seems strange that the losing accumulator was never consulted on a change of choice, i.e. that this change of choice was solely driven by a change of log odds in the winning accumulator.

So the bottom line is that, at least as written, it appears to me that one of two awkward modeling choices is occurring:

A) The decision variable is determined by 2 independent integrators of a decision variable but confidence and changes of decision are determined by an odds ratio that combines the 2 integrators, in which case, why wasn't the initial decision made based upon a measure that combines the two integrators, rather than having this sudden switch between what determines the decision?

B) The log odds are determined only by the winning accumulator's log odds, in which case, it seems odd to have a change of choice that occurs without consideration of the new winner's accumulator.

Finally, there were several important points that do not require more analysis but that the reviewers thought merited serious consideration and discussion:

7) The evidence for separate mechanisms includes a correlation of confidence parameters across subjects when completing distinct tasks that is not seen in the decision-making parameters. The model in this paper seems to be distinct from that – it seems that one would expect the low-high confidence boundaries to remain intact but the decision boundary to change when a new task is performed. Hard to imagine why this would occur in one model, so this issue should at least be discussed.

Perhaps the authors data do explain this, but it is unclear to me – there is a well-known "high-low effect" whereby subjects are more confident than they should be at hard decisions (low coherence) and less confident than they should be at easy decisions (high coherence). My intuition is that if a decision is made yet more information affecting confidence is received post-decision then that extra information would more likely favor the decision (so boost confidence) in easy decisions but as often as not favor the non-chosen target (so reduce conference) in difficult decisions. Is this effect accounted for in the authors' framework? The universal increase in log-odds ratio shown in Figure 7 suggests not (but I have not gone through the calculated effect here so may be wrong).

8) In trying to think through the arguments about the amount of stimulus time not affecting the initial decision, I started wondering about a related time scale that I'm not sure was considered: how long does it take to reverse an already begun motor action? If this takes a long time, then it seems that there may be a very narrow window to actually accomplish a change of mind reflected in a hand movement direction change. Is this accounted for implicitly in the time scale arguments and model for the deadline for change of mind, i.e. can the state of belief at t_pip_ actually turn around the hand this late into the process?

9) I think the authors should mention and compare their model with a recent paper by Wei and Wang (J Neurophysiol) which purports a biophysical mechanism that produces similar results. I think one discrepancy may be the confidence in errors (Figure 2) being of incorrect trend in their model.

With respect to this final point, note that in the reviewer discussion it was made clear that we do not intend you to code up the Wang model, but instead to discuss the likely similarities and differences.

---

## [Author Response]

*Essential revisions: First, there were concerns about the extent to which the common mechanism argument was a complete description of the data Two reviewers raised a related point on these lines. 1) The model fits in* Figure

*2 show a good fit to accuracy and RT data (panels A and B) but a poorer fit to the proportion of high confidence responses (C), particularly for error trials. E.g. subject 1 has only two (out of 6) error-trial data points situated near the fitted line, and only one data point for subject 2 (part of the issue in realising this is that the purple error trial points are often hidden behind the green correct trial points, it would be useful to jitter these sideways for visualisation). This is the case even though there are several hundred error trials particularly at lower coherences with correspondingly tight error bars on confidence proportions. This discrepancy is particularly apparent at 0% coherence, where subjects 1, 2 and 4 show substantially lower confidence than predicted by the model. I'm not suggesting therefore that there can't be a common mechanism that does a good job of explaining all these data features (e.g. perhaps subjects are inferring on coherence, leading to lower confidence at 0% than predicted by a model that marginalizes coherence?). But it seems reasonable that a putative common mechanism driving choice, RT and confidence should do an equally good job of explaining each feature separately.*

The number of parameters in the model is small, by design, and we would like to remind the reviewers that the main focus of the paper is on change of mind about confidence and direction. We have applied more complex models to initial-choice data (e.g., Kiani et al., Neuron, 2014) but elected to go with the simpler model here so as to place emphasis on the paper’s novel contribution. We are aware that the fits are not perfect, and we now acknowledge this in the revised manuscript. The pathology in our view arises from several factors: the fit is driven by the more frequent observations (i.e. correct choices); we do not fit the correlation between races which is fixed to a sensible value (only a few values of correlation lead to an analytically tractable fitting procedure); the bounds are stationary (i.e., flat). We suspect that a combination of these factors contribute to the poorer fit of the errors. We suspect the model may be missing a feature that would be required to explain the confidence at 0% coherence, and we are now forthcoming on this observation.

As requested we have jittered the less visible points in Figure 2.

*2) First, it is not clear to me what data could not be fitted by their model, given multiple fitted parameters per subject. It would be nice to see how a model with a separate "confidence-generating" and "decision-generating" circuit would produce data that would not be fit by their model.*

Examining separate "confidence-generating" and "decision-generating" processes is a useful suggestion, albeit somewhat open ended, since a “confidence-generating” mechanism that does not exploit the information in the decision variable (DV), as ours does, could conceivably exploit a wide class of non-normative sources. We considered several possibilities and now mention two in the paper, in response to the reviewers’ summary point (#4) below.

*3) I do agree though that using Occam's razor, fitting by one model circuit is preferable to two. However, Figure 2 shows a horrible fit of the model to the 0% coherence data points in a systematic direction across subjects. This suggests a systematic problem with the model, but the authors do not (I think) mention this discrepancy.*

We addressed this above in point #1.

*4) There was agreement in the reviewer discussion that simulation of competing models to demonstrate how the data disconfirm a model that does not rely on a common mechanism would be important if the argument is to shift broader opinion. The reviewers regarded such a demonstration as essential. One suggestion during the discussion was as follows: "Perhaps a simplification of their model and then use of Model comparison techniques to adjust for different numbers of parameters would be a good test to show that a separate computation is needed for confidence as they claim."*

As suggested, we have examined two alternative models that do not use a separate computation for confidence (i.e., a criterion in log-odds space). Model 1 exploits a classic idea from signal-detection-theory, which would base confidence solely on the balance of evidence. This information is also used in our model, but the simpler alternative that we now consider ignores elapsed deliberation time. In Model 2, confidence is based on decision time only, thus ignoring the balance of evidence. We fit both models to our participants’ data. They have the same number of parameters as the original model (*df*=4), but the fits are substantially poorer (compared to the original model, log likelihood is reduced by 63-1180 and 2301-3749 across participants for models 1 and 2, respectively). We include two figures (Figure 8 and Figure 9) so that the reviewers can appreciate the inadequacy of these models, but we feel it would distract readers from the focus of the paper – the changes in confidence and choice – were we to include them. The exposition is already too heavy on the initial choices, in our view. We report these new analyses and model comparisons in terms of log likelihoods in the revised paper. Also, please see response to point #6 as to why the initial decision is not terminated on a log-odds bound.

Author response image 1.Fits to alternative model 1 that uses a threshold on the balance of evidence (BoE) to assign confidence and ignores time.Graphs follow the same format as in Figure 2 and display the same data (symbols) and axes. The BoE will tend to be greater with longer times. Therefore, this model produces a qualitative mismatch in the middle column. It also predicts that confidence on errors should decrease as a function of motion strength (magenta curves, 3rd column).**DOI:**
http://dx.doi.org/10.7554/eLife.12192.017

Author response image 2.Fits to alternative model 2 that uses a threshold on decision time to assign confidence (shorter decision times are more confident to match the data) and ignores BoE.The striking failure in the middle column arises because for any criterion, the mean of all decision times greater than this criterion will be large and mean of the times less than this criterion will be small (n.b., the red curve for S3, middle column, is mostly off the graph; only the right end of curve is visible). The other qualitative failures are best appreciated by drawing on intuitions associated with a simpler diffusion model – with symmetric, flat bounds for right and left choices. This is mathematically identical to our competing accumulators but with perfect anti-correlation. In the simpler 1-dimensional model, (1) there is no balance of evidence; (2) the DV is at the level of the upper or lower bound at decision termination, (3) the proportion correct is independent of decision time, and (4) mean RT on correct and error choices are the same for any motion coherence. Based on point 3, we would expect the choice functions in the left column to be identical. They are not exactly the same because our model is a race between two negatively correlated but non-redundant accumulators, and in our particular instantiation (e.g., no reflecting lower bounds), this yields slightly faster errors, which are assigned higher confidence (left column). The same logic applies to the right column.**DOI:**
http://dx.doi.org/10.7554/eLife.12192.018

Related to these concerns, was a concern about data exclusion. 5) This worry is compounded by the exclusion of data that shows aberrant confidence behaviour. One subject was excluded for a strong high-confidence bias

*and absence of confidence-accuracy correlation, which is indeed at odds with the proposed mechanism linking choice and confidence, but perhaps not with other models. Similarly the second half of subject 4's data is excluded because they changed the way they reported confidence half way through the session. These exclusions seem to unfairly skew the playing field in favour of the authors' single-mechanism account, particularly as the remaining subjects' data constitute only a little over half of all data that was originally collected. During the discussion it was made clear that although it is understandable that subjects may have effort-related or other biases that make the data uninteresting, it is a concern when this is 2 of 6 subjects, and it is a concern that the particular exclusions seem to act in your favour. Because of this, it was thought to be essential that the excluded data were presented in the supplement, and that the reason for the odd behaviour was investigated and described.*

We required subjects to have sufficient perceptual skills and motivation to perform the task. One subject was excluded very early as they were at chance on all coherence levels during initial training.

A second subject was excluded because he responded with high confidence on 95% of trials (and with 90% high confidence even at 0% coherence). After 5 sessions (3330 trials) we decided to replace this subject. We found that despite the idiosyncrasy in his data, our model can fit the data quite well. We now show the fit to his initial choices in a new Figure 2—figure supplement 1. Note that because of the low frequency of low confidence decisions, the fits are dominated by the high confidence choices, which are fitted well.

With regard to the subject for whom we only used the first half of the data, no stationary model can account for non-stationary behavior. This subject showed a qualitative change in behavior on the second half of their data (after ~5000 trials), as shown in new Figure 2—figure supplement 2. Importantly, we are able to fit each half of this subject's data separately with our model. We have now included the second half of the data and the model fits in new Figure 2—figure supplement 2.

*6) There was also an important question about perceived asymmetries in model that the reviewers required clarification about. It was unclear why you switch between working in accumulator space and log-odds space. In the discussion, it was agreed that it might be possible for you to clarify in the text if there was a strong and compelling reason, but if not then it would be interesting to see a symmetric model. My largest comment is that, while confidence and decisions are clearly put together in a single framework, I found the framework to be a little awkward and/or unclear in what information/type of model informed decisions versus what informed confidence. The authors use a classic "race to bound" model to inform the decision. However, then they appear to use a separate probabilistic judgment (log odds, or maybe log odds ratio-this was a bit unclear) for confidence and for changes of decision and confidence. This seems awkward: why not just use the log odds metric for all stages of the task instead of artificially making a race to threshold for two completely independent integrators for the decision and then a separate probabilistic framework for confidence and changes of decision?*

*Or, maybe I am misinterpreting the log odds graph and this could simply be interpreted as a one-to-one mapping between the decision variable and the log odds (so that one could indeed reformulate the race to bound in the decision variable as a race to bound in log odds). If this is the case, then I find the log odds measure for change of choice to be somewhat strange: in the race to threshold model, there are two completely independent accumulators that are racing against each other. In the log odds analysis, it would seem very awkward if only the odds for the initially winning accumulator was used (i.e. I'm assuming either the authors are using odds or odds ratios based upon using both accumulators*, *or they are just using the winning accumulator, since the figures only indicate a single trace in log odds). If only the winning accumulator is used, then it seems strange that the losing accumulator was never consulted on a change of choice, i.e. that this change of choice was solely driven by a change of log odds in the winning accumulator.*

*So the bottom line is that, at least as written, it appears to me that one of two awkward modeling choices is occurring:*

*A) The decision variable is determined by 2 independent integrators of a decision variable but confidence and changes of decision are determined by an odds ratio that combines the 2 integrators*,

*in which case, why wasn't the initial decision made based upon a measure that combines the two integrators, rather than having this sudden switch between what determines the decision?*

*B) The log odds are determined only by the winning accumulator's log odds*,

*in which case, it seems odd to have a change of choice that occurs without consideration of the new winner's accumulator.*

We accept the reviewers’ concern about “awkwardness” and will address it in detail after responding to the “bottom line,” which is that neither alternative (A) or (B) characterizes our modeling choice. Beginning with the latter (B), we agree that it would have been odd to ignore the new winner’s accumulator, but that is not what we did. Log-odds is calculated by taking all available information into account, i.e., the state of the winning accumulator, the state of the losing accumulator, and elapsed time. Regarding the first point (A), we actually conceive of similar operations in the initial- and post-initiation epoch. Both rely on the two integrators (which are negatively correlated, so not independent) and elapsed time to establish the log-odds for confidence, and in both epochs, the choice is based on the states of the competing accumulators. We were opaque on this last point because we adopted, and failed to explain, a simplification, applied to the post-initiation epoch only, that allowed us to display the change-of-direction bound as a threshold in belief space. We will justify the simplification in a moment, but the important point is that there is really no conceptual “asymmetry.” Before justifying the simplification, we need to explain the more fundamental awkwardness that the reviewers spotlight – that is, why wasn't the initial decision based upon a measure that combines the two integrators, such as log-odds?

The reason is that such a termination on log-odds is inconsistent with existing behavioral and neurophysiological data. For example, (1) If decisions terminate on the same log odds correct, confidence (by definition) would be the same for all choices, independent of reaction time and stimulus strength. Even if the log-odds bounds were to move in time, the confidence would be purely time-dependent and not depend on coherence (inconsistent with Kiani et al., Neuron, 2014). (2) A constant log-odds correct bound would produce a very flat choice function (probability correct as a function of coherence) and bizarrely long reaction times. (3) Neural representations of accumulated evidence have been reported in several areas of the monkey brain. The drift and noise are consistent with the representation of a DV, and there is a signature of a terminating bound applied to the neural representation of the winning DV. All of the information to compute log-odds correct is available in the state of the competing DVs and elapsed time, but there is no evidence that the firing rate of the competing accumulators combine these sources to represent log-odds explicitly.

Therefore, there is ample justification for assuming different circuits for DV and belief (if belief is explicitly represented at all). Importantly, however, it is easy to show that any computation based on belief (e.g., reporting confidence or its changes) can be formulated as a time dependent criterion on the DVs.

In principle, everything we just stated about the initial decision should apply to the post-initiation epoch to explain change of mind about confidence and/or direction. However, we made a simplifying assumption that allowed us to perform both calculations in belief space. Because the information available in this epoch is limited by *t_nd_* and motor costs, we assumed that subjects would base their choices on all the available information up to some point (*t_pip_*≤ *t_nd_*). Thus we are ignoring explicitly (it is now explicit in the revision) the likelihood that a change of mind has a termination rule. We think it does (with all kinds of interesting motor costs), but there are too few trials to achieve traction on this, as we now explain. Expressing changes of confidence via a criterion in log-odds is already the natural choice, of course. We conceive of the change of mind about direction as operating on the state of the DVs at *t_pip_*. For example, were the criterion for change of choice simply a change in the sign of the balance of evidence, this would be identical to a log-odds criterion at zero. However, the hysteresis effect for choice places the criterion on DV on one or the other side of zero. In summary, our simplifying assumption introduces a minor approximation which allows us to illustrate changes of confidence and choice on the same figure, which we view as useful to readers – or should be now that we have cleared up the confusion.

Finally, we acknowledge that it is possible to view both confidence and choice as operating on DVs and elapsed time. This is the approach that yields the optimal policy (stopping rule) for deciding among the four options (left/right direction x high/low confidence) based on optimizing the points per unit time (or some other desideratum), as pursued by Drugowitsch and Rao groups, for example.

We have clarified the approximation and our rationale in the revised manuscript (see Methods).

*Finally, there were several important points that do not require more analysis but that the reviewers thought merited serious consideration and discussion. 7) The evidence for separate mechanisms includes a correlation of confidence parameters across subjects when completing distinct tasks that is not seen in the decision-making parameters. The model in this paper seems to be distinct from that* –

*it seems that one would expect the low-high confidence boundaries to remain intact but the decision boundary to change when a new task is performed. Hard to imagine why this would occur in one model, so this issue should at least be discussed.*

The reviewers seem to be making an intriguing connection to a controversial literature bearing on the question of whether individuals performing different tasks tend to exhibit consistent “metacognitive” sensitivities, such that some individuals are consistently better than others at discriminating correct from incorrect decisions (e.g., Fleming et al., Science 2010). We do not have much to say about this, but the analysis in Figure 7 demonstrates one reason why a subject might have higher metacognitive sensitivity than another. When confidence is reported after a choice, confidence can be influenced by additional evidence accrued after the commitment to a choice. The idea is similar to one discussed by Pleskac and Busemeyer (2010), and more recently by Moran, Teodorescu and Usher (2015) and Yu, Pleskac and Zeigenfuse (2015). The analysis of Figure 7 demonstrates that the opportunity to revise the initial confidence based on additional evidence leads to higher ‘metacognitive’ sensitivity. Therefore, an intriguing possibility is that individuals with higher metacognitive sensitivity are those that can accumulate evidence for longer periods of time (or more efficiently) after the commitment to a choice. This mechanism could account for individual differences in sensitivity without the need to postulate separate mechanisms for choice and for confidence. Furthermore, it is empirically testable, with analyses like those shown in Figure 4. We now have added the following to Discussion:

“The evaluation of additional evidence may explain why confidence ratings are more strongly correlated with accuracy when they are reported with less time pressure (Yu et al., 2015), and why some individuals appear to be better than others at discriminating correct from incorrect decisions (e.g. Ais et al., 2016).”

*Perhaps the authors data do explain this, but it is unclear to me* –

*there is a well-known "high-low effect" whereby subjects are more confident than they should be at hard decisions (low coherence) and less confident than they should be at easy decisions (high coherence). My intuition is that if a decision is made yet more information affecting confidence is received post-decision then that extra information would more likely favor the decision (so boost confidence) in easy decisions but as often as not favor the non-chosen target (so reduce conference) in difficult decisions. Is this effect accounted for in the authors' framework? The universal increase in log-odds ratio shown in Figure 7 suggests not (but I have not gone through the calculated effect here so may be wrong).*

We don’t need to resort to changes of mind to explain the high-low effect. For example, Kiani et al. (Neuron 2014) showed that even at the lowest coherence, where participants are at chance, the predicted certainty is around 0.7. And for the high coherence trials certainty is at around 0.9 despite almost perfect performance. So the high-low effect arises naturally from the diffusion model. The reviewers’ intuitions are partly correct, but the deterministic component of the DV contributes a biastoward higher confidence for all correct trials. The deeper intuition behind high-low is to think about all the motion strengths and directions that could have given rise to the DVs at decision time. The DVs achieved in a high-coherence trial could occasionally arise in a low- or intermediate-coherence trial. This possibility biases confidence to lower values despite nearly perfect accuracy on high-coherence trials. Similarly, the DVs achieved in a low-coherence trial could be occasionally observed in an intermediate- or high-coherence trial, biasing confidence on low-coherence trials to higher values despite near chance accuracy.

*8) In trying to think through the arguments about the amount of stimulus time not affecting the initial decision, I started wondering about a related time scale that I'm not sure was considered: how long does it take to reverse an already begun motor action? If this takes a long time, then it seems that there may be a very narrow window to actually accomplish a change of mind reflected in a hand movement direction change. Is this accounted for implicitly in the time scale arguments and model for the deadline for change of mind, i.e. can the state of belief at t_pip_ actually turn around the hand this late into the process?*

Yes, this is implicitly included in *t_pip_*, but we were less than clear on this, and thank the reviewers for bringing this to our attention. We suggest that *t_pip_* (Figure 3) is the amount of the non-decision time participants will use after the initial decision to determine whether to change one’s mind. This is partially determined by the physical effort to change the trajectory of the hand. We showed previously (Burk et al., Plos One, 2014) that manipulating the effort required to change one’s mind can affect the number of changes of mind and *t_pip_*. We revised the legend to Figure 3 to make the latencies clearer, and we reinforce the concept that *t_pip_* cannot exceed *t_nd_*.

9) I think the authors should mention and compare their model with a recent paper by Wei and Wang (J Neurophysiol) which purports a biophysical mechanism that produces similar results. I think one discrepancy may be the confidence in errors (Figure

2C) being of incorrect trend in their model. With respect to this final point, note that in the reviewer discussion it was made clear that we do not intend you to code up the Wang model, but instead to discuss the likely similarities and differences.

The Wei and Wang paper focuses mainly on the post-decision wagering task (Kiani & Shadlen, 2009), although they devote a section to the RT version of the task. Wei and Wang estimated confidence based only on the difference of neural activities in a ring attractor model. Computation of confidence in their model is in essence similar to the Balance of Evidence model explained above (see point #4). It suffers from the same shortcomings too. For example, as the reviewers mentioned, the model predicts that confidence in errors decreases with stimulus strength, contrary to the data. Despite this failure, we appreciate Wei and Wang’s model as a step toward understanding the circuit that underlies choice and confidence. They are aware of the discrepancy between their model and experimental results and are working toward a better solution (see their Discussion).

We now cite this paper when we mention the alternative model that bases confidence solely on balance of evidence (i.e., ignoring elapsed time).